# A chromosome-level genome sequence of *Chrysanthemum seticuspe*, a model species for hexaploid cultivated chrysanthemum

Michiharu Nakano [1], Hideki Hirakawa[2], Eigo Fukai [3], Atsushi Toyoda [4], Rei Kajitani [5], Yohei Minakuchi[4], Takehiko Itoh [5], Yohei Higuchi[6], Toshiaki Kozuka[1], Hidemasa Bono [1], Kenta Shirasawa [2], Ippei Shiraiwa[1], Katsuhiko Sumitomo[7], Tamotsu Hisamatsu[7], Michio Shibata[6], Sachiko Isobe [2], Kenji Taniguchi[1] & Makoto Kusaba [1✉]

Chrysanthemums are one of the most industrially important cut flowers worldwide. However, their segmental allopolyploidy and self-incompatibility have prevented the application of genetic analysis and modern breeding strategies. We thus developed a model strain, Gojo-0 (*Chrysanthemum seticuspe*), which is a diploid and self-compatible pure line. Here, we present the 3.05 Gb chromosome-level reference genome sequence, which covered 97% of the *C. seticuspe* genome. The genome contained more than 80% interspersed repeats, of which retrotransposons accounted for 72%. We identified recent segmental duplication and retrotransposon expansion in *C. seticuspe*, contributing to arelatively large genome size. Furthermore, we identified a retrotransposon family, SbdRT, which was enriched in gene-dense genome regions and had experienced a very recent transposition burst. We also demonstrated that the chromosome-level genome sequence facilitates positional cloning in *C. seticuspe*. The genome sequence obtained here can greatly contribute as a reference for chrysanthemum in front-line breeding including genome editing.

[1] Graduate School of Integrated Sciences for Life, Hiroshima University, Higashi-Hiroshima, Hiroshima, Japan. [2] Kazusa DNA Research Institute, Kisarazu, Chiba, Japan. [3] Graduate School of Science and Technology, Niigata University, Niigata, Niigata, Japan. [4] National Institute of Genetics, Mishima, Shizuoka, Japan. [5] School of Life Science and Technology, Tokyo Institute of Technology, Meguro-ku, Tokyo, Japan. [6] Graduate School of Agricultural and Life Sciences, The University of Tokyo, Bunkyo-ku, Tokyo, Japan. [7] Institute of Floricultural Science, National Agriculture and Food Research Organization, Tsukuba, Ibaraki, Japan. ✉email: akusaba@hiroshima-u.ac.jp

Asteraceae is one of the largest angiosperm families, comprising one-tenth of all angiosperm species. Species within this family grow across various habitats on all continents except for Antarctica[1]. They are characterized by unique floral/fruit traits that may contribute to the evolutionary and ecological success of Asteraceae; for instance, the capitulum, a single flower-like structure composed of a number of florets, may attract insects, whereas the pappus, a modified calyx, may promote seed dispersal[2].

Chrysanthemums (*Chrysanthemum morifolium* Ramat.) rank second in the world cut flower market after roses[3]. Chrysanthemums have enchanted people worldwide because of their variety in attractive flower colors and morphologies (Fig. 1). Typical chrysanthemums have capitula consisting of florets with short petals, which form a central disc; ray florets with long decorative petals at the outermost layer; and bracts, which are modified leaves mimicking calyxes[4]. Some chrysanthemums have multiple-layered ray florets, which sometimes have fused tube-like ligules (Fig. 1). Combinations of such floral morphological traits, together with color variation, create visual complexity of the chrysanthemum capitulum, which is the result of accumulated mutations selected by ancient to present-day breeders.

Wild species in the genus *Chrysanthemum* are classified into four groups, the *indicum* group, *makinoi* group, *zawadskii* group, and *Ajania* group, according to their morphological characteristics and molecular phylogenetic analysis[5]. Wild chrysanthemum species have capitula similar to that of "wild-type" *C. morifolium*, except for the *Ajania* group. Each group contains polyploid species from diploids up to decaploids, which is an interesting evolutionary characteristic of the genus *Chrysanthemum*[6]. Another characteristic is the occurrence of interspecific hybridization even between species with different polyploidy levels[7]. Notably, the cultivated chrysanthemum is thought to originate from interspecific hybrids involving the *indicum* group species and others[7–9].

Plants with high polyploidy levels tend to be larger in sizes[10]. The majority of cultivated chrysanthemums are segmental allohexaploids, which may result in large capitula and elevated ornamental value; however, this complicates genetic analysis[11,12]. Furthermore, self-incompatibility in the cultivated chrysanthemum leads to a highly heterozygous state of the genome[13]. These factors have prevented the application of modern breeding systems used for other crops, and the prevailing strategy of chrysanthemum breeding is still based on simple crosses and clonal propagation using stem cuttings.

Gojo-0, a pure line bred from the self-compatible mutant of *C. seticuspe* (Maxim.) Hand.-Mazz.—a diploid species ($2n = 18$) belonging to the *indicum* group—was developed as a model strain of the genus *Chrysanthemum*[5]. Because it is diploid and

self-compatible, Gojo-0 can be used for screening recessive mutants. Gojo-0 completes its life cycle twice a year in a growth chamber, and *Agrobacterium*-mediated leaf disc transformation is feasible. Nevertheless, it is essential to determine whole-genome sequence information for use as a model strain.

In this study, we obtained a high-quality, whole-genome sequence of Gojo-0 on the chromosome-level, which revealed characteristics of the genome structure and evolution of *C. seticuspe*, such as recent retrotransposon expansion and segmental duplication. Furthermore, we demonstrate that the chromosome-level sequence information supports the value of Gojo-0 as a model strain of the genus *Chrysanthemum*, and can contribute to the development of new traits that have not been utilized in the breeding of cultivated chrysanthemums, such as self-compatibility.

## Results and discussion

**Assembly and annotation of *C. seticuspe* genome sequence.** To obtain the chromosome-level, whole-genome sequence of Gojo-0, we generated 317.4 Gb (~99× coverage) and 343.0 Gb (~107× coverage) of data on the Illumina HiSeq 2500 and PacBio Sequel systems, respectively. These short- and long-read sequences were assembled using the Platanus-allee v2.2.2 assembler[14] and then gap-closed using the assembled result with long read sequences by the Flye assembler[15], resulting in 13,928 scaffolds (Table 1). Based on the distribution of *k*-mer frequencies, the estimated genome size of Gojo-0 was 3.15 Gb (Supplementary Fig. 1), which is slightly larger than that of *C. seticuspe* line XMRS10 (3.06 Gb), a Gojo-0 sibling whose draft genome sequence is available[16]. Next, 396 million read pairs were obtained from the Hi-C library by HiSeq sequencing, after which scaffolding of the assembled sequences was performed using the HiRise method (Supplementary Figure 2). The total length of the resultant assembled sequence was 3.05 Gb, which covers 97% of the Gojo-0 genome (Table 1). The largest nine scaffolds, which corresponded to the haploid chromosome number of *C. seticuspe*, accounted for 95% (2.99 Gb) of the Gojo-0 genome, with the longest and shortest scaffolds measuring 373 and 263 Mb, respectively (Supplementary Table 1). The nine scaffolds were numbered as linkage groups (LGs) to correspond with those of *C. morifolium*[17].

A total of 632,376 genes were initially predicted on the repeat masked Gojo-0 genome by BRAKER2[18], after which 620,134 genes were extracted as "best" by excluding the variants constructed *in silico* (Supplementary Fig. 3). According to our criteria, the best genes were classified into 71,916 high confidence (HC) genes, 258,832 low confidence (LC) genes, and 289,386 transposable elements (TEs). A total of 58,555 high-quality Iso-Seq sequences were generated from the Iso-Seq reads derived from four SMRT cells. The high-quality sequences were collapsed

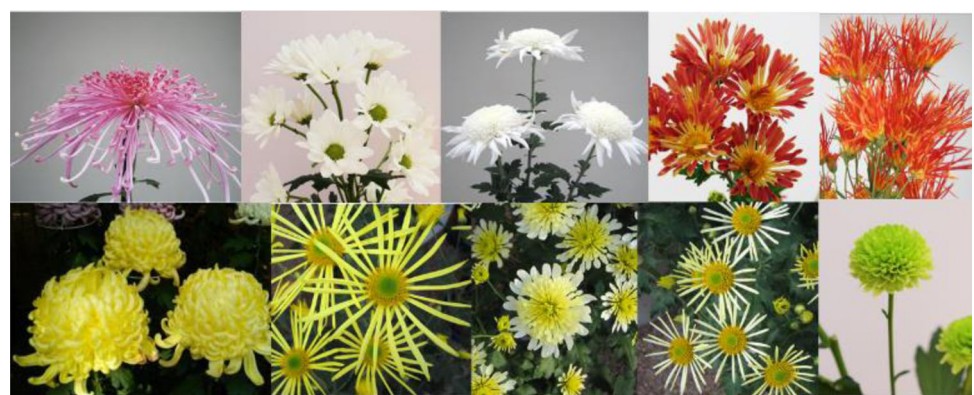

**Fig. 1** Morphological and color variation among chrysanthemum cultivars.

**Table 1 Statistics of the Gojo-0 genome assembly.**

|  | Platanus/Flye | Gojo-0_v1 |
|---|---|---|
| Total length | 3042.49 Mb | 3,045.06 Mb |
| LG1-9 | – | 2,988.17 Mb |
| Number of scaffolds | 13,928 | 8,493 |
| Scaffolds > 10 kb | 5996 | 902 |
| Scaffolds > 100 kb | 4162 | 40 |
| Scaffolds > 1 Mb | 937 | 9 |
| Percent of N | 1.95% | 1.97% |
| Percent of GC | – | 35.5% |
| Scaffold N50 | 1.027 Mb | 347.131 Mb |
| Number of genes | – | 74,259 |
| BUSCO complete % (embryohyta_odb10) | – | 97.4% |
| LAI / raw LAI | – | 18.32/23.78 |

into 32,775 sequences, and 14,415 complete confidence (cc) sequences were identified using ANGEL. The cc sequences were classified into 14,325 HC genes and 90 TE genes. The HC and TE genes were then mapped onto the repeat masked genome sequence by GMAP[19] and replaced with the genes predicted by BRAKER2[18]. LC genes that were similar to the protein sequences of Araport11 were manually confirmed, and those with significant hits were reclassified as HC. In addition, 11 known genes that were not predicted by BRAKER2[18] and Iso-Seq data were added to the HC group. Finally, we obtained 74,259 HC; 251,308 LC; and 282,975 TE genes (Table 1). The HC sequences were regarded as intrinsic protein-coding sequences and were subjected to further analysis. The predicted gene number was comparable to that of XMRS10 (71,057). To evaluate completeness of the assembled sequence, BUSCO[20] analysis was performed and the resultant values were 97.4% for the whole genome, suggesting that the quality of assembly was sufficiently high. In addition, long terminal repeat (LTR) Assembly Index of the Gojo-0 genome was 18.32, which also guaranteed high quality of the genome sequence[21] (Table 1).

**Genome evolution and expansion in *C. seticuspe*.** Although *C. seticuspe* is a diploid species, it has a relatively large genome (3.15 Gb) and a large number of predicted genes (74,259) compared with those of well-established model plants such as *Arabidopsis thaliana* L. (0.13 Gb; 27,655 genes)[22]; this may represent a possible minimum gene set. To address this, we investigated the structure and evolution of the *C. seticuspe* genome. The phylogenetic tree constructed using whole-genome sequences suggests that in the tribe Anthemideae, *C. seticuspe* and *C. nankingense* diverged approximately 3.1 million years ago (Mya) and diverged from *Artemisia annua* L. approximately 6.1 Mya (Fig. 2a)[23–25]. *Erigeron canadensis* (Astereae) is thought to have diverged from Anthemideae approximately 28 Mya[26], and lettuce (Cichorieae)[26] is more distantly related to *Chrysanthemum* than to sunflowers (Hellantheae)[27] and *Mikania micrantha* Kunth (Eupatorieae)[28], which was consistent with previous studies[2].

The Asteraceae family experienced whole-genome triplication (WGT-1) 38–50 Mya after diversification of Asterid I, which includes coffee plant (*Coffea canephora*) and Asterid II, which includes Asteraceae species (Fig. 2a). Plot analysis of the distribution of synonymous substitutions per site (Ks) between Gojo-0 pseudochromosomes indicated a peak at Ks = 0.8 (Fig. 2b). Peaks at the same position were observed between lettuce pseudochromosomes and between *E. canadensis* pseudochromosomes, which corresponded to WGT-1. Dot plot analysis of *C. seticuspe* pseudochromosomes illustrated the three paralogous regions in the *C. seticuspe* genome, which are shared by

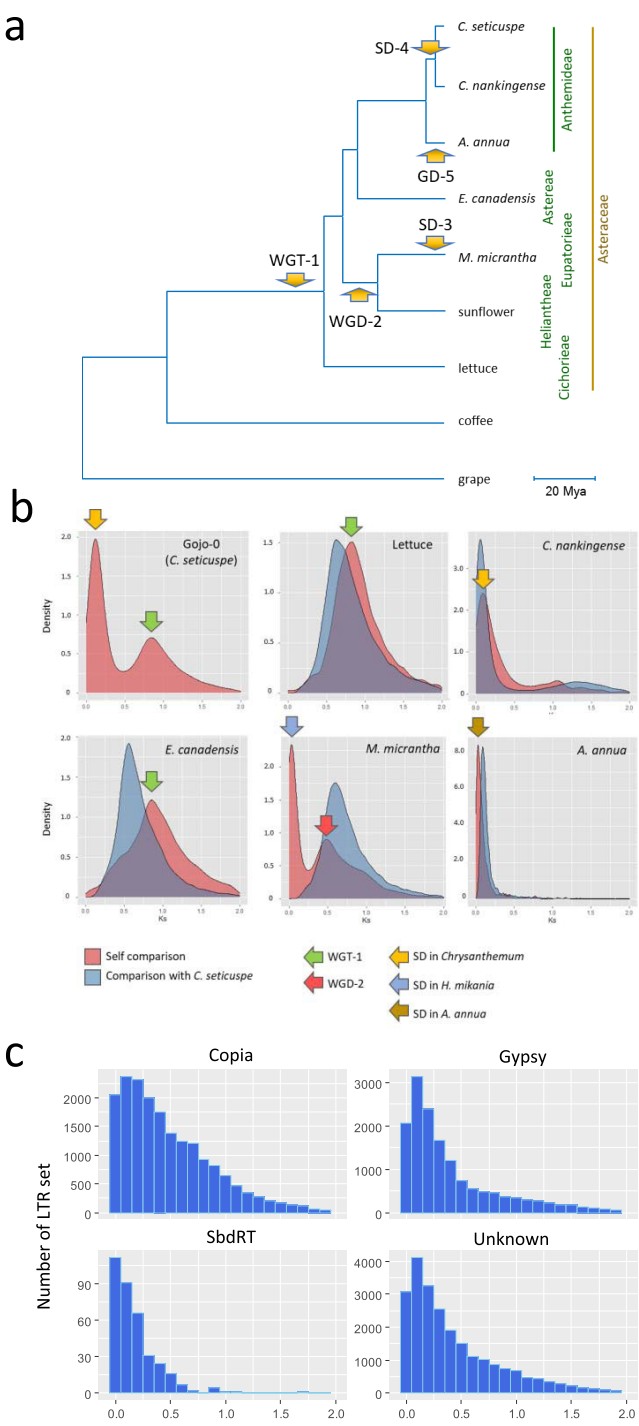

**Fig. 2 Genome evolution of *Chrysanthemum seticuspe*. a** Phylogenetic tree of Asteraceae species. Grape and coffee plants[78] were used as the outgroups. Arrows indicate ages of the whole-genome triplication (WGT-1) event common in Asteraceae; whole genome duplication (WGD-2) before diversification between *M. micrantha* and *H. annuus*; segmental duplications in *M. micrantha* (SD-3); before diversification between *C. seticuspe* and *C. nankingense* (SD-4); and in *A. annua* (GD-5). Mya, million years ago. **b** Distribution of synonymous substitutions per site (Ks). Pink and blue charts indicate plots of self-comparison and comparison with *C. seticuspe*, the peaks of which represent speciation time from *C. seticuspe*. **c** Distribution of the insertion time of intact long terminal repeat (LTR) retrotransposons in *C. seticuspe*.

grape (*Vitis vinifera* L.)[29], *E. canadensis*, and lettuce, confirming that *C. seticuspe* experienced whole genome duplication before speciation (Supplementary Fig. 4). Interestingly, *C. seticuspe* showed another peak at Ks = 0.1, which was not present for lettuce and *E. canadensis*. Moreover, long stretches of syntenic blocks were not frequently observed in the dot plot analysis of *C. seticuspe* pseudochromosomes, although the predicted duplication at Ks = 0.1 occurred recently (Fig. 2b; Supplementary Fig. 4a). Considering that the Ks plot was constructed using genes with microsynteny, the results suggest that this duplication is a segmental duplication (SD-4) rather than a duplication involving the whole genome. The same peak was observed in the self-comparison of the *C. nankingense* genome, which has a larger Ks value than that of the *C. seticuspe-C. nankingense* speciation (Fig. 2b), suggesting that SD-4 occurred before speciation of *C. seticuspe* and *C. nankingense*. Meanwhile, the Ks value of the peak observed in the self-comparison of *Artemisia annua* was smaller than that of SD-4, suggesting that SD-4 is not shared by the genera *Chrysanthemum* and *Artemisia* and that it occurred after speciation. Recently, it was reported that *Mikania micrantha*, an invasive species belonging to Asteraceae, experienced a very recent segmental duplication (SD-3) event, which is thought to contribute to rapid environmental adaptation (Fig. 2a)[28].

In general, genome duplications, transposition of retroelements, and an increase in repeat sequences contribute to genome expansion in plants[30]. As commonly observed in plant species with large genome sizes, inter-spread repeats occupied a large portion (80.3%) of the whole genome length of *C. seticuspe* (Table 2). Long terminal repeat retrotransposons (LTR-RTs) accounted for 72% of interspersed repeats (57.6% of the whole genome length), and included Copia-type (40.1%) and Gypsy-type (31.4%) LTR-RTs. Among the intact LTR-RTs retaining both 5′ and 3′ LTRs, the most abundant family in the Copia and Gypsy superfamilies were SIRE and Athila, respectively (only intact LTR-RTs were analyzed hereafter; Supplementary Table 2). SIRE and Athila are the most abundant families in *Tanacetum cinerariifolium* and *A. annua*, both belonging to tribe Anthemideae together with *C. seticuspe*[31]. Insertion time analysis revealed that the Copia and Gypsy superfamilies as well as unknown-type LTR-RTs were most frequently transposed approximately 0.2 Mya (Fig. 2c), indicating that retrotransposon expansion occurred very recently in *C. seticuspe*. The transposition burst is thought to have occurred around 1 Mya in *C. nankingense*, suggesting that it occurred independently after speciation of *C. seticuspe* and *C. nankingense*, which was predicted to have occurred approximately 3.1 Mya (Fig. 2a)[23]. Meanwhile, it is possible that the same LTR-RT family may have contributed to genome expansion in

Anthemideae because the most abundant LTR-RTs are common among Anthemideae species[31].

**Characterization of a novel LTR-RT: SbdRT.** We isolated a flower morphology mutant of *C. seticuspe*, *shiboridama* (*sbd*), from a progeny of the natural population (Fig. 3a, b). *sbd* produced shoot-like structures instead of florets on the receptacle of the capitulum—the single flower-like inflorescence characteristic of Asteraceae. The phenotype of floret to shoot conversion is similar to that of loss-of-function mutants of the floral meristem identity gene *LEAFY*, suggesting that the *LEAFY* ortholog in *C. seticuspe* (*CsFL*) may be impaired in *sbd* (Fig. 3c)[32,33]; indeed, DNA sequencing revealed an ~8 kb insertion in the second exon of *CsFL* (Fig. 3d). *sbd* was found to be a single recessive mutant with a phenotype that is perfectly linked to the *CsFL* genotype in 48 segregating individuals (Fig. 3e). These observations suggest that *CsFL* is a likely candidate gene responsible for the *sbd* phenotype. The 8 kb insertion has 775 bp direct repeats at both ends, indicating that this insertion is an LTR-RT (Fig. 4a, Supplementary Data 1a, Supplementary Data 2). The severe deleteriousness of *sbd* (infertility) suggests that the mutation has not been retained for long during evolution and that this insertion was recently inserted into *CsFL*. Indeed, the 5′- and 3′-LTR sequences, which are identical in active LTR-RTs, are identical in this insertion (Supplementary Data 1a, Supplementary Data 2). We named this insertion 'Shiboridama Retrotransposon-nonautonomous insertion-original' (SbdRT-nis-ori).

SbdRT-nis-ori was thought to have transposed recently, but the internal 3.9 kb sequence showed no similarity to any other LTR-RTs and contained no valid open reading frame (ORF), indicating that it is a nonautonomous retrotransposon (Fig. 4a). Therefore, we speculated that there are autonomous copies of the SbdRT family in the *C. seticuspe* genome that provides trans-activation components for the transposition of SbdRT-nis-ori. We tried to identify SbdRT family copies with LTRs sharing more than 90% identity with those in SbdRT-nis-ori in the Gojo-0 genome and identified 360 sequences (Supplementary Data 3). Among the 360 sequences, 161 were found to encode all or some of the functional domains that are characteristic of an LTR-RT ORF, i.e., gag-polypeptide of LTR Copia-type, GAG-pre-integrase domain, integrase core domain, and transcriptase (RNA-dependent DNA polymerase) domain (Fig. 4a, Supplementary Table 3). These 161 copies were classified as SbdRT-orf. Among the SbdRT-orf copies, 67 had a long ORF containing all four domains, suggesting that they are autonomous copies of SbdRT that can activate SbdRT-nis-ori in trans (Fig. 4a). The encoded amino acid sequence and the order of the functional domains also suggest that SbdRT belongs to the Copia superfamily[34]. Most SbdRT-orfs were assigned to the Angela clade (Supplementary Table 3).

A closer examination of the protein-coding sequence revealed two SbdRT-orf groups (Fig. 4a); the ORF lengths in group 1 were 4155 or 4158 bp while those in group 2 were 4071 or 4074 bp. Group 2 showed a T insertion in the ORF corresponding to the position between the 66th and 67th nucleotide of group 1 ORF, causing a short N-terminal truncation. SbdRT-nis-ori completely lost the long ORF and instead contained a 3,861 bp-long unknown noncoding internal sequence, designated as "nis" (Fig. 4a). The majority of non-protein-coding SbdRT copies (190/199) contained the nis sequence and were therefore classified as SbdRT-nis. Even though SbdRT-orf and SbdRT-nis were easily distinguishable in the ORF/nis region, the rest of the sequences in addition to the LTRs were similar. The region between 5′-LTR and ORF/nis (PBS_ATG) shared 91.9% identity between SbdRT-nis-ori and SbdRT-orf (CsG_LG5_64792825-64801000; Fig. 4a; Supplementary Data 3). The AT-rich 3′ region

### Table 2 Repeat composition in Gojo-0.

|  | Number of elements | Length occupied | % of sequence |
|---|---|---|---|
| Total length (LG1-9) | - | 2,988,175,699 | 100 |
| Based masked | - | 2,425,091,008 | 81.2 |
| Retroelements | 1,812,817 | 1,746,847,923 | 58.5 |
| LINEs: | 44,166 | 26,497,323 | 0.9 |
| LTR elements | 1,768,651 | 1,720,350,600 | 57.6 |
| DNA transposons | 63,503 | 44,440,749 | 1.5 |
| Rolling-circles | 10,409 | 6,960,574 | 0.2 |
| Unclassified | 1,168,814 | 608,286,289 | 20.4 |
| Total interspersed repeats | - | 2,399,574,961 | 80.3 |
| Simple repeats | 303,610 | 16,883,708 | 0.6 |
| Low complexity | 34,095 | 1671,765 | 0.1 |

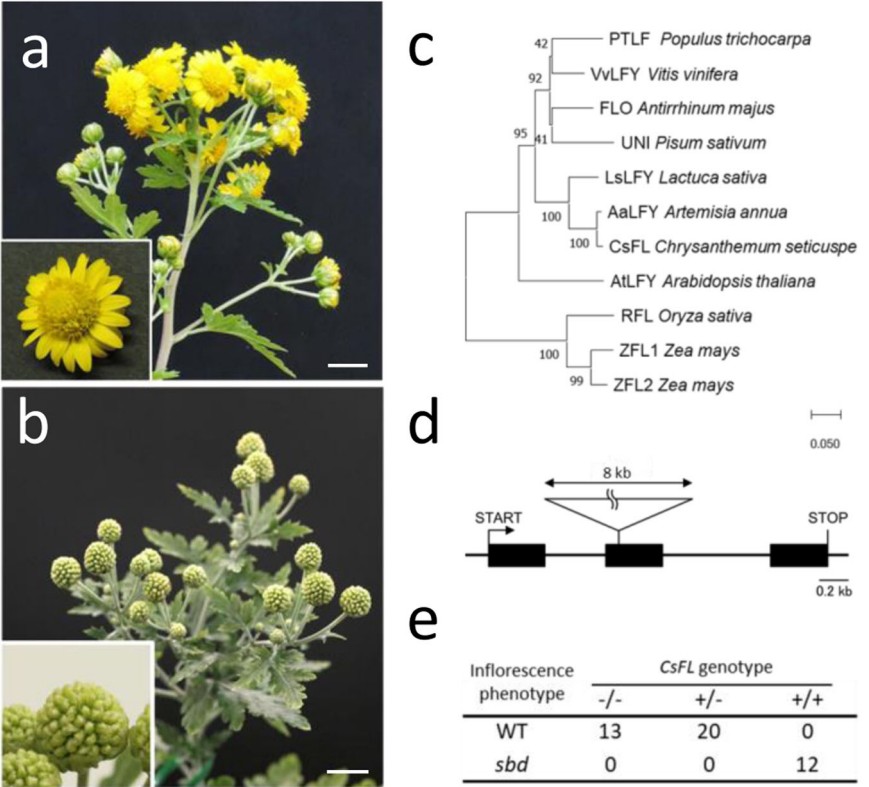

**Fig. 3 *SHIBORIDAMA* is the *LEAFY* ortholog in *C. seticuspe*. a** Capitula of wild-type (WT) *C. seticuspe*. **b** Organs corresponding to the capitula in the *shiboridama* (*sbd*) mutant. **c** Phylogenetic tree of *LEAFY* orthologs. *CsFL* is the *LEAFY* ortholog in *C. seticuspe*. Most *LEAFY* orthologs are single copies, with the exception of recently genome-duplicated species such as maize[79]. **d** Structure of *CsFL* in *sbd* mutant. **e** Linkage analysis between the inflorescence phenotype and *CsFL* genotype. + and – indicate with and without insertion, respectively. Bars indicate 1 cm.

downstream of ORF/nis (3_ATrich) appeared polymorphic, with 75.0% identity between SbdRT-nis-ori and SbdRT-orf. The region between 3_ATrich and 3'-LTR (3_cons) showed a high level of identity (94.0%; Fig. 4a). This high-level conservation of the non-coding regions suggests their functional importance for transposition in *cis* and *trans*. In addition, the conserved SbdRT-nis copies occupied 52.7% of SbdRT copies (190 out of 360), suggesting that SbdRT was found to be an LTR-RT that is unique as nonautonomous type is highly prosperous (Supplementary Table 3).

Interestingly, 360 SbdRT copies were mainly localized in the distal regions of chromosomes, unlike intact Copia and Gypsy superfamilies (Fig. 4b). As is often observed in plants with large genome sizes, a high gene density was observed in the distal regions of chromosomes in *C. seticuspe*. Interestingly, distribution of predicted genes closely corresponded to that of SbdRTs in Gojo-0 pseudochromosomes (Fig. 4b). In fact, the ratio of SbdRT within or flanking structural genes was higher than that of general intact-LTR-RTs (Supplementary Figure 5). Estimation of insertion time of SbdRT suggests that they transposed very recently (peaked < 0.1 Mya), which is earlier than for other LTR-RTs in *C. seticuspe* (peaked ≈ 0.2 Mya; Fig. 2c), raising the possibility that SbdRTs retain high transposing activity.

Investigation of the distribution of SbdRT among plant species revealed that both SbdRT-orf and SbdRT-nis were present in *C. nankingense*, implying that they diverged before speciation (Fig. 4c; Supplementary Data 1a, b, Supplementary Data 2). Notably, *C. seticuspe* SbdRTs had shorter branches than *C. nankingense* SbdRTs in the phylogenetic tree, suggesting that SbdRTs transposed and were amplified very recently in *C. seticuspe*.

**Characterization of genes related to floral development.** Chrysanthemum cultivars exhibit various capitula morphologies (Fig. 1), which are generated by the accumulation of mutations in floral development. Therefore, it is important to elucidate capitulum development to design capitulum structures for chrysanthemum breeding. The wild-type chrysanthemum capitulum is composed of two types of florets: radially symmetrical tubular florets and bilaterally symmetrical ray florets[4]. Tubular florets at the center are bisexual with short equivalent petals, whereas the ray florets at the margin are female with long ventral petals that form a showy ligule. In contrast, the capitulum of lettuce consists of only ray florets, whereas the artichoke and *A. annua* capitula develop only tubular florets. The transcription factor gene *CYCLOIDEA* (*CYC*)—first identified in *Antirrhinum majus*—determines the bilateral symmetry of flowers, and recent studies on several Asteraceae species have reported that *CYC2* family genes are involved in determining floral symmetry and inflorescence architecture[35–39]. Indeed, we identified six *CYC2* family genes in the Gojo-0 genome (Supplementary Fig. 6, Supplementary Data 4). Phylogenetic analysis of the *CYC2* family genes suggested that there are four clades (CsCYC2a, CsCYC2c, CsCYC2d, and CsCYC2e/f classes) in Asteraceae, and *C. morifolium*, *C. seticuspe*, gerbera (*Gerbera hybrida*), and sunflower that have both tubular and ray florets in a capitulum retain all four classes of *CYC2*. Interestingly, lettuce, artichoke[40], and *A. annua*, which form discoid or ligulate homogamous capitula, lack one of the four classes of *CYC2*. It has been suggested that the relatively large number of *CYC2* family genes in Asteraceae is a consequence of genome triplication (WGT-1) and contributes to the complexity of the capitulum[26,41]. Our analysis confirmed that *CYC2* family genes are located in the syntenic regions of

 **5**

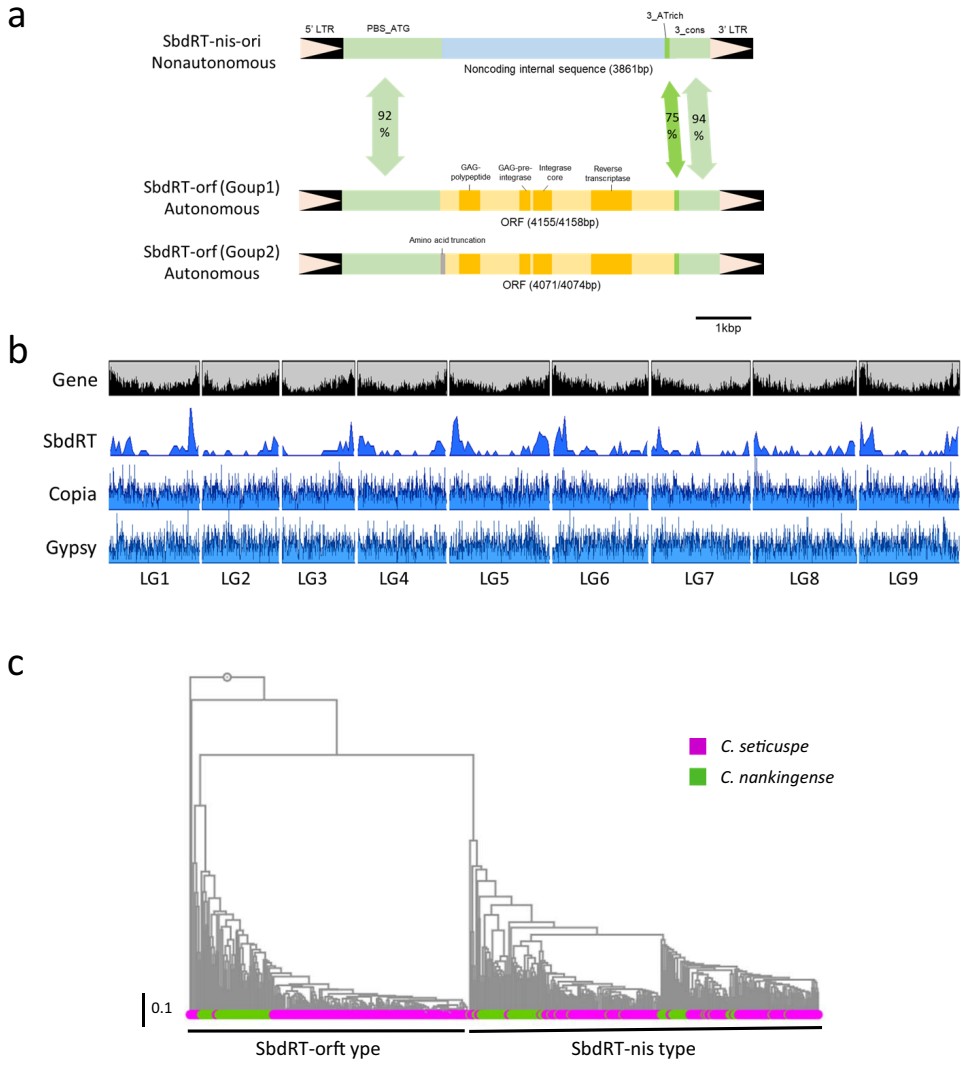

**Fig. 4 Structure and distribution of *shiboridama*-retrotransposon. a** Structures of SbdRT in the *C. seticuspe* genome. SbdRT-nis-ori is the original SbdRT found in *CsFL* of the *sbd* mutant, which carries a noncoding insert instead of the long open reading frame (ORF) in SbdRT-orf. **b** Distribution of intact long terminal repeat-RTs in the pseudochromosomes of Gojo-0. Densities of predicted genes (high confidence), SbdRT copies of all types, and members of the Copia and Gypsy superfamilies are shown along with the nine linkage groups. **c** A phylogenetic tree of SbdRT copies in *C. seticuspe* and *C. nankingense*. Only the full-length SbdRT copies were subjected to analysis.

*C. seticuspe*, lettuce, and *E. canadensis*, although some *CYC2* family genes are absent in lettuce and *E. canadensis* (Supplementary Fig. 4). Interestingly, some *CYC2* family genes were found located in a limited region (in LG4 in *C. seticuspe* and lettuce, and in LG2 in *E. canadensis*), suggesting that tandem duplication also contributes to amplification of the *CYC2* family genes in Asteraceae (Supplementary Fig. 4).

In the analysis of ABCE-class MADS-box genes, which are essential for floral organ identity[42], we found two genes each in the *SEP3*, *SEP1/2*, *SEP4*, *FUL*, and *AG* clades, one gene each in the *AGL6*, *AP1/CAL*, and *PI* clades, and three genes in the *AP3* clade in the *C. seticuspe* genome. In addition, we identified a clade containing two *C. seticuspe* MADS-box genes without counterparts in *A. thaliana* and two *C. seticuspe* STK-like genes (Supplementary Fig. 7, Supplementary Data 4). The number of paralogous genes in each clade was similar among *C. seticuspe*, lettuce, artichoke, and *A. annua*, except that the *AP3* clade gene was present as a single copy in lettuce but as three copies in the other species. Such a distribution of ABCE-class MADS-box genes suggests that these genes do not have a major role in tubular/ray floret differentiation.

In cultivated chrysanthemum, six *CYC2*-like genes (*CmCYC2a/b/c/d/e/f*) have been reported, and overexpression of *CmCYC2c* in *C. lavandulifolium* promotes the petal growth of ray florets[38]. Phylogenetic analysis showed that each *CYC2* paralog in *C. morifolium* had a close counterpart in *C. seticuspe* (except for *CmCYC2b*), confirming a strong relationship between the species as well as the usefulness of Gojo-0 as a reference for cultivated chrysanthemum (Supplementary Fig. 7).

**Positional cloning in *C. seticuspe* using the chromosome-level genome sequence.** Positional cloning strategies for molecular cloning have not been applied to cultivated chrysanthemums as they exhibit segmental alloploidy and self-incompatibility. To demonstrate the usefulness of chromosome-level, whole-genome sequence information, we attempted positional cloning of the *ALBINO1* (*ALB1*) gene of *C. seticuspe* as an exemplar. *alb1* is a single recessive mutation segregated in the selfed progeny of AEV02 (Fig. 5a)[5].

We first selected 6370 expressed sequence tag-derived simple sequence repeat (EST-SSR) markers with amplicons shorter than

# a

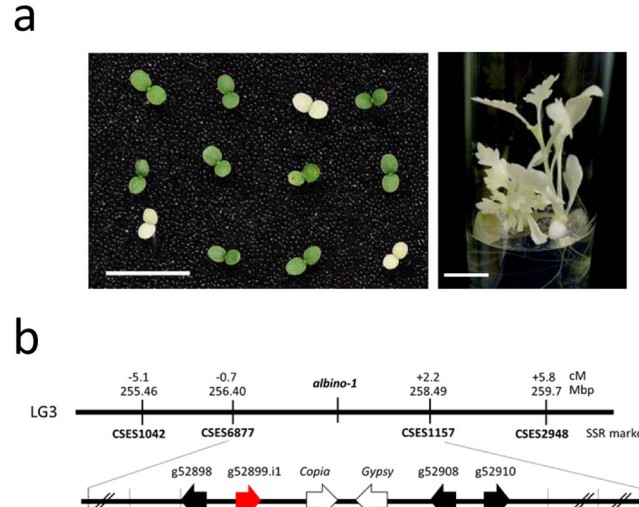

# b

# c

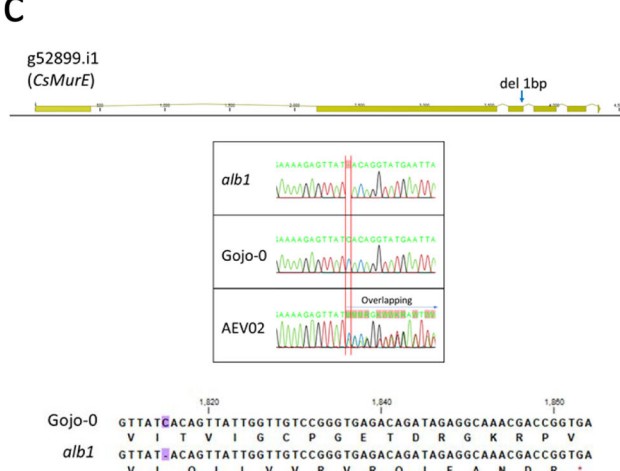

**Fig. 5 Positional cloning of *ALB1* in *C. seticuspe*. a** Phenotype of *alb1*. Left, two-week-old seedlings of selfed AEV02 progeny segregating wild-type (green) and *alb1* mutant (white) phenotypes. Right, in vitro culture of *alb1*. Scale bar indicates 1 cm. **b** Structure of the *ALB1* candidate region. Analysis of the F2 population revealed that *ALB1* was located between SNP markers 257.216 and 255.277. **c** Structure of *CsMurE* in *alb1*. A single base pair deletion was observed in the third exon (heterozygous in AEV02) resulting in a frameshift and premature stop codon.

1000 bp and mapped them to the Gojo-0 pseudochromosomes (Supplementary Fig. 8, Supplementary Data 5)[16]. The markers were approximately evenly distributed on the nine pseudochromosomes, suggesting that the SSR markers are useful for genetic mapping. Next, we performed genetic mapping of *ALB1* using the selfed AEV02 progeny. Initial mapping using 23 albino individuals located *ALB1* between SSR makers CSES6877 (256.4 Mb) and CSES1157 (258.49 Mb) on LG3 (Fig. 5b). To design more dense DNA markers, we determined the DNA sequences of genes predicted in the candidate region, focusing on introns to detect single nucleotide polymorphisms (SNPs). Using such SNP markers and 968 albino segregants, we delimited the candidate region to ~61 kb between the markers SNP257.216 and SNP257.277. In this region, four genes and two RTs were predicted.

Among these four genes, g52899.i1 was found to encode MurE ligase, which is also present in bacteria and conserved among plant species[43]. g52899.i1 shares 65.3% amino acid sequence identity with MurE ligase in *A. thaliana* (AtMurE) and retains the functional domains observed in AtMurE, such as the transit peptide for chloroplast transition and Mur ligase domains (Supplementary Fig. 9). Syntenic analysis between g52899.i1 in LG3 of *C. seticuspe* and *AtMurE* on chromosome 1 of *A. thaliana* confirmed microsynteny between them, suggesting that g52899.i1 is the ortholog of *AtMurE*, and was thus designated *CsMurE* (Supplementary Fig. 10). Interestingly, the loss-of-function mutant of *AtMurE* shows an albino phenotype[43]. *alb1* carries a single base deletion in the third exon, resulting in a frameshift, which deletes most of the C-terminal domain of MurE ligase, eliminating its function (Fig. 5c, Supplementary Fig. 10). There was another MurE-like sequence in Gojo-0/AEV02 (corresponding to a part of LG6.g04440.1), but it lacked substantial similarity, particularly in the 5′ region, and the translated amino acid sequence contained premature stop codons, suggesting that it is a pseudogene (Supplementary Fig. 11). Therefore, *C. seticuspe* has a single functional *MurE* in its genome, and MurE activity is completely impaired in *alb1*, which is thought to cause the albino phenotype. These observations indicate that *ALB1* encodes CsMurE. Similar to *AtMurE*, *CsMurE* was expressed at high levels in leaves, confirming that CsMurE plays an important role in chloroplast development (Supplementary Fig. 12).

**Utilizing the Gojo-0 genome information in hexaploid chrysanthemum breeding**. It has long been proposed that cultivated chrysanthemum is derived from interspecific hybrids involving the *indicum* group[8,9,44]. Therefore, *C. seticuspe*, a member of the *indicum* group, has similar properties to cultivated chrysanthemum even though *C. seticuspe* may not be a direct ancestor. Consistent with this, *C. seticuspe* genes were found to share high similarities with their orthologs in cultivated chrysanthemum; for example, most orthologs of flower morphology/flowering-related genes share more than 98% nucleotide sequence identity (Supplementary Data 6). Thus, Gojo-0 may be a useful reference for cultivated chrysanthemum research. Indeed, *C. seticuspe* has been used as a model for cultivated chrysanthemum in the analysis of photoperiod/temperature-regulated flowering time[45,46], seasonal dormancy[16], flower color, and self-incompatibility[5].

The large genome size of *C. morifolium* (for example, that of the cultivar 'Sei-marine' was estimated at 7.9 Gb with relative fluorescence intensity to Gojo-0; Supplementary Fig. 13) and its autopolyploid-like nature make it difficult to obtain a high-quality, whole-genome sequence, which is important as a tool for modern DNA sequence-based breeding. The DNA marker-based linkage map of cultivated chrysanthemum very closely corresponded to the genome sequence of Gojo-0 (Supplementary Fig. 14); this suggests that the chromosome-level genome sequence of Gojo-0 can be informative for trait mapping of cultivated chrysanthemum and is a promising reference for the construction of chromosome-level, whole-genome sequences of *C. morifolium*. Thus, lessons from genetic studies in *C. seticuspe* can be applied to genetic analysis/breeding of cultivated chrysanthemum, even though cultivated chrysanthemum have three sets of homologous chromosomes.

Unlike alloploid plants, functional diversification of "homeologous" genes has not progressed extensively in autoploid and segmental allopolyploid species partly due to frequent exchange of genetic information between genes on different homologous chromosomes. For example, in the segmental allohexaploid *C. morifolium*, six genes at the same locus on different homologous chromosomes are known to be randomly inherited (hexasomic

inheritance), implying that all six genes must be "homozygous" for recessive alleles to show the complete recessive phenotype[12]. In fact, haplotype analysis of the *CsFL* transcripts in the cultivar 'Jinbudiao' revealed that at least four different haplotypes (alleles) exist in an individual, suggesting that *C. morifolium* retains at least four alleles at the single *CsFL* locus, which reflects the recent hexaploid formation (Supplementary Fig. 15). In the diploid self-compatible strain, a recessive phenotype emerges with a probability of 25% in the selfed progeny when the parent plant has one recessive mutant allele. In *C. morifolium*, it takes at least three generations to obtain the recessive mutant phenotype when the parent plant carries one recessive mutation. Given that *C. morifolium* is a self-incompatible species, it is extremely difficult to obtain individuals in which all six genes are homozygous for the recessive allele. Therefore, the traditional forward-genetic breeding system in chrysanthemum is considered to have failed to utilize useful mutations without obvious phenotypes unless the six genes are homozygous for the recessive allele.

As shown in this study, positional cloning in Gojo-0 can be performed similar to that in other model plants. Isolation of mutants and identification of their responsible genes in Gojo-0 may lead to the discovery of useful novel genes that have not been utilized in chrysanthemum breeding. Whole-genome sequence information is also useful for quantitative trait locus (QTL) or genome-wide association study (GWAS) cloning, which facilitates maker-assisted selection (MAS). In addition, Gojo-0 probably harbors active retrotransposons, which can be used for gene tagging if appropriate conditions for induction of transposition are found[47,48]. Furthermore, reverse genetic breeding is a prospective breeding strategy for crops with a high-level autoploidy-like nature, such as cultivated chrysanthemum. We can select individuals with useful combinations of alleles using DNA markers identified in QTL or GWAS analyses (i.e., MAS) even in the F1 cross populations during breeding of self-incompatible chrysanthemum. In particular, it is noteworthy that all six homologous genes can be reportedly knocked out efficiently in the hexaploid chrysanthemum[49], suggesting that results obtained from genetic analysis using Gojo-0 can be reconstituted in a reverse genetic manner in cultivated chrysanthemum.

Although a wide variation in flower color and morphology is observed in cultivated chrysanthemum (Fig. 1), a number of useful recessive mutations are thought to remain unutilized in the current breeding programs because of the high-level segmental alloploidy and self-incompatibility. The chromosome-level whole-genome sequence and plant resources of Gojo-0 can contribute to the analysis of interesting traits in the family Asteraceae and genus *Chrysanthemum* as well as facilitate modern breeding programs based on MAS and genome editing that focus on useful novel genes, such as those involved in self-incompatibility and axillary lateral bud outgrowth in the cultivated chrysanthemum.

## Methods

**De novo genome assembly and Hi-C scaffolding**. Genomic DNA was extracted from the young leaves of *C. seticuspe* Gojo-0 line (10th selfed generation) using the Cetyl trimethyl ammonium bromide (CTAB) method[50] and QIAGEN Genomic-tips (Qiagen, Hilden, Germany). A paired-end library and four mate libraries were constructed using a TruSeq DNA PCR-Free Library Prep Kit and a Nextera Mate Pair Sample Preparation Kit (Illumina, San Diego, CA), respectively. The final libraries were sequenced using the Illumina HiSeq 2500 System in rapid mode. For long-read sequencing, an SMRTbell library was constructed using an SMRTbell Express Template Prep Kit (Pacific Biosciences, Menlo Park, CA) according to the manufacturer's protocol. The sequencing library was size-selected using the Blue-Pippin system (Sage Science, Beverly, MA) with a minimum fragment length cutoff of 30 kb. The 32 SMRT cells were run on the PacBio Sequel system with Sequel Binding Kit 3.0 and Sequel Sequencing Kit 3.0 (Pacific Biosciences). Short and long read sequences were assembled using the Platanus-allee v2.2.2 assembler[14]. Gap

closure was performed using the assembled result with Platanus-allee assembled sequences by the Flye assembler[15]. The sequences (Platanus/Flye, Table 1) were then subjected to Hi-C proximity-ligation (Dovetail Genomics, Scotts Valley, CA). In the Hi-C analysis, paired-end sequences were obtained using the Hiseq X System (Illumina), after which assembly was performed using the HiRise software pipeline (Dovetail Genomics). Next, polishing was performed using SMRT Link v8.0.0.80529 (Pacific Biosciences; arrow: defaults) and Pilon v1.22 (–fix indels)[51].

The genome size of *C. seticuspe* Gojo-0 was estimated by *k*-mer analysis using Jellyfish[52] (*k* = 27) and GenomeScope[53]. The quality of the genome assembly was estimated using BUSCO v4.0.5[20], with embryophyta_odb10 (1614 BUSCOs).

**Gene prediction and annotation**. Iso-Seq sequences were generated from the shoot tips, leaves, axillary buds, flower buds, and flowers of the *C. seticuspe* accession NIFS-3. Total RNA was extracted using RNAiso Plus (Takara Bio, Shiga, Japan), followed by the RNeasy Mini Kit (Qiagen) according to manufacturers' instructions. Libraries were constructed using the SMRTbell Template Prep Kit 1.0 SPv3 or SMRTbell Express Template Prep Kit 2.0 (Pacific Biosciences), and sequences were obtained using a PacBio Sequel System with four SMRT cells (Sequel SMRT Cell 1 M v3 LR; Pacific Biosciences). The isoforms were generated by Iso-Seq2 (SMRT Link v5.1) or Iso-Seq3 (SMRT Link v6) for sequences derived from each of the two SMRT cells.

Gene prediction of the Gojo-0 assembled genome sequence was performed as described in Supplementary Fig. 3. Repetitive sequences were searched using RepeatModeler v1.0.11 (http://www.repeatmasker.org) and Repbase v23.05[54], and nucleotide sequences without similarities against protein sequences were searched using DIAMOND v0.9.29[55] in more sensitive mode. The resultant repetitive sequences were soft-masked using RepeatMasker v4.0.7. Gene prediction was performed for the repeat masked sequences using the RNA-Seq data of *C. seticuspe* (DRX080967-DRX081001) and the protein sequence of *C. nankingense* (v2.0) obtained from the Chrysanthemum Genome Database (http://www.amwayabrc.com) with BRAKER v2.1.5[19] (Supplementary Fig. 3). Genes with the highest score among variants at each gene locus were selected and termed "best" genes. The best genes were classified into three categories (HC, LC, and TE) based on (1) similarity searches against the two databases UniProtKB (https://www.uniprot.org/) and NCBI NR (https://www.ncbi.nlm.nih.gov/refseq) using DIAMOND in sensitive mode with *E*-value≤1e$^{-80}$ and identity ≥ 80%; (2) BLASTP v2.8.0[56] search against the two protein sequences of *C. seticuspe* (CSE_r1.1; https://plantgarden.jp/ja/list/t111766/genome/t1111766.G001) and *C. nankingense* (v2.0; http://www.amwayabrc.com) with *E*-value≤1e$^{-80}$, identity≥90%, and 60%≤length coverage≤140%; (3) protein family search against Pfam 31.0 (http://pfam.xfam.org/) using HMMER v3.2.1 (http://hmmer.org/) with *E*-value≤1e$^{-50}$; and (4) expression analysis for transcripts per million (TPM) by Salmon v1.2.1[57]; genes with TPM > 0.0 were considered expressed. Genes showing similarities to TEs or whose product names were related to TEs according to UniProtKB, were classified as TE. The remaining genes fulfilling (1) to (4) above were classified as HC, while other genes were classified as LC. High-quality, full-length consensus isoforms were constructed using Iso-Seq3 on SMRT Link v9 (Supplementary Fig. 3). High-quality sequences were then collapsed on the Gojo-0 assembled genome sequence using the Tofu pipeline in Cupcake. ORFs were predicted using ANGEL against the collapsed filtered sequences, and those with start and stop codons were selected as cc sequences. The cc sequences were classified into HC and TE as described in Supplementary Fig. 3. The HC and TE genes derived from Iso-Seq data were mapped onto the Gojo-0 genome sequence using GMAP v2020.06.01[19]. Genes in a sense direction at the splice site, longest genes in each gene locus, and genes mapped in length from 95 to 105% were selected, whereas those less than 99 nt in length were excluded. If genes were predicted by BRAKER2 and Iso-Seq sequencing at the same gene locus, those supported by Iso-Seq were preferentially selected. The sequences classified as LC were further subjected to a BLASTP search against the protein sequences of Araport11[22] with *E*-value≤ 1e$^{-10}$, length≥60%, coverage≤140%, and identity≥25%; sequences showing significant similarity were re-classified as HC.

**Ks analysis and whole genomic duplication analysis**. Pairs of orthologous genes were identified using MCscanX[58]. The protein sequences used in this study were downloaded from public databases; *C. nankingense* (v2.0, http://www.amwayabrc.com/download.htm), *A. annua* (assembly accession: GCA_003112345.1), *E. canadensis* (V1, https://genomevolution.org/coge/GenomeInfo.pl?gid=56884), sunflower (*H. annuus;* GCF_002127325.2), lettuce (*L. sativa;* GCA_002870075.2), *M. micrantha* (GCA_009363875.1), coffee (*Coffea canephora;* v1.0, https://coffee-genome.org/download), and grape (*V. vinifera;* Genoscope.12X). The protein sequences were compared with those of *C. seticuspe* Gojo-0 using BLASTP with a threshold of 1e-10. The BLASTP results were fed into MCscanX using gene position data obtained from the gff file of the genome information of each species with default settings. The syntenic relationship was illustrated by Synvisio[59] (https://synvisio.github.io/#/). The "add_kak-s_to_synteny.pl" MCscanX plugin was used to calculate Ks values for each gene pair. The data obtained were summarized using the dplyer and ggplot2 packages of R studio v1.4.1106 with R v4.0.4. Finally, 11 genes, which are included in the gene set of XMRS10 but not that of Gojo-0, were added to HC gene set and 29 duplicated genes were removed from HC.

**Repeat annotation**. The *de novo* repeat libraries were developed for whole-genome data using RepeatModeler v2.0.1[60] and LTR_retriever v2.9.0[21]. For RepeatModeler, a repeat library was obtained using the default parameters. For LTR transposons, the resulting outputs of the LTR harvest[61] and LTR Finder[62] were fed into the LTR_retriever program to develop both the repeat library and intact LTR-RT catalog, after which the quantitative metric LAI was calculated. The repeat libraries were used as repeat databases for summarizing repeat frequency using Repeat-Masker v4.1.1. Intact LTR-RTs were then classified into families as follows: superfamilies were classified as Copia, Gypsy, or unknown based on the LTR retriever analysis results. Intact LTR-RTs defined by LTR_retriever were classified using the DANTE tool at the Galaxy server (https://repeatexplorer-elixir.cerit-sc.cz/galaxy/)[63]. The DANTE classification results for protein domains were counted for each intact LTR-RT; family names with a frequency > 0.5 and matching the superfamily classification of LTR_retriever were used as the class of each retrotransposon. The relative positions of the coding region of LTR-RTs were evaluated using snpEFF[64] with the HC annotation gene set. The distribution of LTR transposons was illustrated using KalyoploteR[65].

**Identification and characterization of SbdRT family members in the Gojo-0 genome**. To identify SbdRTs in the Gojo-0 genome, we first conducted a BLASTN search[56] using a 775 bp LTR of SbdRT-nis-ori as a query. Homologous sequences showing homology≥90% with coverage≥70% of the query were selected as candidates for LTRs. Next, pairs of two tandemly arranged LTR candidates within 2–10 kb intervals were identified as SbdRT family sequences. The positional information of the identified 360 SbdRT family copies in the Gojo-0 genome is listed in Supplementary Data 3. The longest ORFs were then predicted from each copy, and the predicted amino acid sequences were subjected to Pfam[66]. The 360 SbdRT copies were classified as SbdRT-orf/group1, SbdRT-orf/group2, SbdRT-nis, and others according to Supplementary Data 3 (see also Results and Discussion section).

Representative SbdRTs were selected from each clade of a phylogenetic tree and aligned using the CLC main workbench v21.0.2 (Qiagen) with ClustalW in the plugin "Additional Alignments."

**Phylogenetic tree construction and divergence time prediction**. Orthofinder v2.5.2[67] was used to identify members of gene families and assess the phylogenetic relationships among species. The protein sequences used for the Ks plot were subjected to analysis using Orthofinder with default settings. The divergence time was calculated using "make_ultrametric.py" in Orthofinder, with the divergence time between grapes and *Chrysanthemum* set to 118 Mya according to TimeTree[68]. The phylogenetic tree of the *LEAFY*, *CYC2*, and ABCE-class MADS-box genes was constructed according to the amino acid sequences of each protein using the neighbor-joining method in MEGA X[69]. Bootstrap values for 500 resamplings are shown for each branch.

To construct a phylogenetic tree of SbdRT copies in *C. seticuspe* and *C. nankingense*, a catalog of *C. nankingense* intact LTR-RTs was generated by LTR_retriever v2.9.0[21], after which the fasta sequences were extracted using Samtools faidx[70]. The LTR sequence of SbdRT-nis-ori was subjected to a BLASTN search against the *C. nankingense* intact LTR-RT sequences, and homologous sequences were selected using the same parameters employed for Gojo-0 genome analysis. Full-length LTR-RTs, including the 145 *C. nankingense* SbdRTs and 360 Gojo-0 sbdRTs, were aligned using MAFFT with default parameters online[71]. The phylogenetic tree was reconstructed using UPGMA and viewed with Archaeopteryx.js (https://github.com/cmzmasek/archaeopteryx-js).

**Estimation of insertion time for LTR-RTs**. The 5′- and 3′-LTRs from each copy of LTR-RTs were aligned using ClustalW[72], and Kimura's distance between the two LTRs was calculated using MEGA X[69]. The mutation rate 1.3e-8 bp/site/year was used to calculate the age of the identified LTR-RTs.

**Analysis of the *sbd* mutant**. The *sbd* mutant (AKW12) was discovered in the progeny of a natural *C. seticuspe* population collected in Fukushima, Japan. Linkage analysis was performed using F1 progeny between cross-compatible individuals heterozygous for *sbd*. The primers used for SbdRT-nis-ori sequencing and *CsFL* genotyping are listed in Supplementary Data 7. The PCR products were directly sequenced and assembled using CLC Main Workbench for full-length characterization.

**Characterization of floral morphology genes**. Genes related to flowering and capitulum morphogenesis in *C. seticuspe* were identified based on BLAST searches (see below). In Supplementary Data 4, the nucleic acid sequences of previously reported floral organ-related genes of *C. seticuspe* XRMS10[16] were used as queries for BLASTN search against CsGojo-0_v1.LG0-9_cds_HC_rev2, CsGojo-0_v1.LG0-9_cds_LC_rev2, and CsGojo-0_v1.LG0-9_cds_TE_rev2, or BLASTX search against the NCBI database (https://blast.ncbi.nlm.nih.gov/Blast.cgi).

**Positional cloning of *ALB1***. The selfed progeny of AEV02 were used to map the *ALB1* locus. Previously described SSR marker sets[16] were mapped onto the Gojo-0_v1 genome with Burrows-Wheeler Aligner (bwa aln) and alignments were generated with bwa sampe[73] (Supplementary Data 5). Among the markers mapped on the Gojo-0_v1 genome, a total of 56 SSR primer sets covering the whole genome regions were used for coarse mapping of *ALB1* using bulked DNA from 23 albino individuals in the segregating population (Supplementary Data 5), confining the candidate region within a 20 Mb genomic region. To delimit the candidate region, 968 albino individuals were subjected to further mapping using additional SSR, indel, and SNP markers, which were genotyped with Qsep fragment analyzer (BiOptic Inc., New Taipei City, Taiwan), agarose gel electrophoresis, or direct Sanger sequencing (Supplementary Data 7). Expression of *CsMurE* in various tissues of Gojo-0 and *alb1* was analyzed by qRT-PCR as follows: total RNA was isolated using the Isoplant RNA Kit (Nippon Gene, Tokyo, Japan) and cDNA was synthesized using ReverTra Ace qPCR RT Master Mix with gDNA Remover (Toyobo, Osaka, Japan). PCR was then performed using THUNDERBIRD SYBR qPCR Mix (Toyobo) on a Rotor-Gene Q real-time cycler (Qiagen). The primer set used is listed in Supplementary Data 7 and samples were tested in triplicate per tissue.

**Haplotype analysis**. From the Sequence Read Archive (SRA) database, three sequence run reads in SRA Project ID SRP109613[74] were retrieved using the SRA Toolkit (v2.9.6) (https://github.com/ncbi/sra-tools). For trimming and quality control of reads from the SRA database, Trim Galore! (v0.6.6) (https://www.bioinformatics.babraham.ac.uk/projects/trim_galore) with Cutadapt (v1.18)[75] was applied to filter reads with insufficient quality. The trimmed RNA-seq data were mapped to the Gojo-0 reference genome using the HC annotation gene set with HISAT2 v2.1.0[76]. Alignments to the reference genome were visualized using the Integrated Genome Viewer[77] after data conversion by Samtools[70].

**Ploidy analysis**. The nuclear DNA content of *C. seticuspe* Gojo-0 and *C. morifolium* var. 'Sei-marine' was measured with a Quantum P Ploidy analyzer (Quantum Analysis GmbH) according to the manufacturer's instruction. Briefly, a small piece of fresh leaf was roughly hashed in a staining solution containing DAPI, after which the solution was filtered and applied to the ploidy analyzer.

**Statistics and Reproducibility**. In Supplemental Figs. 12 and 13, we used $n = 3$ biologically independent samples.

**Reporting summary**. Further information on experimental design is available in the Nature Research Reporting Summary linked to this paper.

## Data availability

Raw datasets were deposited in the DDBJ database under Project ID PRJDB7468 for genomic sequencing and PRJDB5536 for Iso-seq. All sequence datasets, assembled genome sequences, predicted gene models, annotations, and high-quality, full-length consensus isoforms are available in PlantGarden (https://plantgarden.jp/; Supplementary Table 4). The genome assembly is also available from the DNA Data Bank of Japan (DDBJ) with the accession numbers BPTQ01000001 to BPTQ01008495 (8495 sequences) and GenBank assembly accession GCA_019973895.1. The plant materials used in this study and their related information are available from the National BioResource Project (https://shigen.nig.ac.jp/chrysanthemum/).

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

## Acknowledgements

We thank Yumi Nagashima for her technical assistance. This work was supported by MEXT KAKENHI (grant number JP18K05619 to M.N.) and a grant from the National Bioresource Project (NBRP) to M.K. and A.T. Some chrysanthemum cultivars (shown in Fig. 1) were provided by Inochio-Seikoen (Japan).

## Author contributions

M.K. and M.N. conceived the project and designed the experiments. H.H., A.T., R.K., Y.M., T.I., K.Sh., and S.I. conducted genomic sequence analysis. M.N. and E.F. conducted retrotransposon analysis. H.B., K.S., T.H., H.H., S.I., and K.Su. conducted expression analysis. Y.H. and M.N. conducted gene family analysis. T.K., I.S., and K.T. analyzed the *shiboridama* mutant. M.N. performed other analyses. M.N., H.H., E.F., A.T., Y.H., H.B., K.Sh., M.S., S.I., and M.K. wrote the paper.

## Competing interests

The authors declare no competing interests.
