## [Peer Review File · Communications Biology]

Reviewers' comments:

Reviewer #1 (Remarks to the Author):

The authors reported a chromosome-level genome sequence of *Chrysanthemum seticuspe*. As the authors stated, Gojo-0, which was sequenced, can be used as a model strain for hexaploid cultivated chrysanthemum. I agree with the authors that the genome sequence will greatly contribute as a reference for chrysanthemum in breeding and research. Therefore, I support the publication of the genome. However, some issues should be addressed before the publication of the manuscript.

Major comments

Point 1: In the title and abstract, the authors emphasized that cultivated chrysanthemum is autohexaploid, but the authors also claimed "the cultivated chrysanthemum is thought to originate from interspecific hybrids involving the indicum group species and others" in Introduction. How the authors proved that cultivated chrysanthemum is autopolyloid?

Point 2: The authors estimated the genome size by using GenomeScope, but different results will appear when using different k-mer in GenomeScope. Please provide the parameters and complete GenomeScope results.

Point 3: It is suggested that LAI should be used to evaluate the quality of genome.

Point 4: How did the authors determine that SD-3 / SD-5 event is segmental duplication but not WGD?

Point 5: How many populations screen to obtain the sbd mutant? Any special treatment? How to locate the candidate gene of the mutant as LEAFY?

Point 6: The assembled genome sequences of this version are not available in PlantGarden (<https://plantgarden.jp/>).

Minor comments

Line 56 *Chrysanthemum* should be italics.

Line 138 Should be Fig. 2 and Supplementary 4a?

Line 198 nonautonomous should be nonautonomous.

Line 334 *chrysanthemum* should not be italics.

Reviewer #2 (Remarks to the Author):

The manuscript presented a chromosome-level genome sequence for a pure line Gojo-0 of *Chrysanthemum seticuspe*. The manuscript is generally well-prepared, and the genome will help understand evolutionary of *Chrysanthemum* species and to gene location for useful trait, thus providing reference for genetic basis of important traits in cultivated chrysanthemum. I suggest publication if the following concerns are satisfactorily addressed.

Major comments

1>Autohexaploid or allohexaploid for cultivated chrysanthemum remains controversial. It is not suitable to determine its nature without sufficient data.

2> *Chrysanthemum* is commonly regarded as cultivated chrysanthemum. Thus far there is no evidence for the origination of cultivated chrysanthemum from *C.seticuspe*. It is not persuasive to claim *C. seticuspe* pure line Gojo-0 as a model chrysanthemum.

3>Cultivated chrysanthemum is so high heterozygous that most traits in F1 crosses with contrasting backgrounds showed different genetic variation. The genome of pure line *C. seticuspe* Gojo-0 is undoubtedly useful, but it difficult to provide reference for the breeding of heterozygous cultivated chrysanthemums. It should be adequately discussed.

In addition, there are many format errors or typos in manuscript and reference list.

Reviewers' comments:

Reviewer #1 (Remarks to the Author):

The authors reported a chromosome-level genome sequence of *Chrysanthemum seticuspe*. As the authors stated, Gojo-0, which was sequenced, can be used as a model strain for hexaploid cultivated chrysanthemum. I agree with the authors that the genome sequence will greatly contribute as a reference for chrysanthemum in breeding and research. Therefore, I support the publication of the genome. However, some issues should be addressed before the publication of the manuscript.

Major comments

Point 1: In the title and abstract, the authors emphasized that cultivated chrysanthemum is autohexaploid, but the authors also claimed “the cultivated chrysanthemum is thought to originate from interspecific hybrids involving the indicum group species and others“ in Introduction. How the authors proved that cultivated chrysanthemum is autopolyploid?

Response: We have revised autohexaploid to hexaploid in the title and to segmental allopolyploid in the abstract and text. Klie et al. (Front. Plant Sci. 5, 479, 2014) and Van Geest et al. (BCM Genomics 18, 585, 2017) claimed that cultivated chrysanthemum is a segmental hexaploid, which is an allohexaploid (originates from interspecific hybrids), but its chromosomes behave like an autohexaploid (all six homologous chromosomes are very similar to each other and all homologous chromosomes can form a pair with each other).

Point 2: The authors estimated the genome size by using GenomeScope, but different results will appear when using different k-mer in GenomeScope. Please provide the parameters and complete GenomeScope results.

Response: We apologize for not clarifying, and have accordingly provided the complete GenomeScope results in Supplementary Fig. 1. The k-mer size used in this analysis was 27.

Point 3: It is suggested that LAI should be used to evaluate the quality of genome.

Response: Thank you for the suggestion. The LAI value was 18.32, which indicates that the quality of the Gojo-0 genome sequence is sufficiently high. We have included this in the text and Table 1.

Point 4: How did the authors determine that SD-3 / SD-5 event is segmental duplication but not WGD?

Response: Liu et al. (Nat. Commun., 11, 340, 2020) revealed that segmental duplication occurs in the evolution of *Mikania micrantha* (SD-3) during their analysis of the *Mikania* genome. Our analysis of the Ks plot (Fig. 2) suggests that genome duplication recently occurred in *Artemisia annua*. However, as you pointed out, it is not easy to conclude that this was segmental duplication because the genome sequence of *A. annua* is not at the chromosome level. Therefore, we revised the expression of 'SD-5' to 'GD-5' (genome duplication-5) in Fig. 2a and the text.

Point 5: How many populations screen to obtain the *sbd* mutant? Any special treatment? How to locate the candidate gene of the mutant as *LEAFY*?

Response: The *sbd* mutant is a spontaneous mutant discovered in a natural population, and thus we did not perform any screening. In addition, we have not performed positional cloning. We assumed that *SBD* is a *LEAFY* ortholog because the phenotype of *sbd* is very similar to that of *leafy* mutants in other species. *sbd* has a large insertion in the second exon of *LEAFY* ortholog in *C. seticuspe* (*CsFL*), suggesting that *CsFL* in *sbd* lost its function completely. The genome sequence revealed that *CsFL* is a single copy gene in *C. seticuspe* genome, suggesting that *sbd* does not have a functional *LEAFY* in its genome. Finally, linkage analysis revealed that the phenotype and genotype perfectly matched in 48 individuals of the segregating F2 population. Thus, we concluded that *CsFL* is the very likely candidate of *SBD*.

Point 6: The assembled genome sequences of this version are not available in PlantGarden (<https://plantgarden.jp/>).

Response: We disclosed the sequence from PlantGarden during review.

Minor comments

Point 7: Line 56 *Chrysanthemum* should be italics.

Point 8: Line 138 Should be Fig. 2 and Supplementary 4a?

Point 9: Line 198 nonautonomous should be nonautonomous.

Point 10: Line 334 *chrysanthemum* should not be italics.

Response: We have made the necessary revisions as suggested.

Reviewer #2 (Remarks to the Author):

The manuscript presented a chromosome-level genome sequence for a pure line Gojo-0 of *Chrysanthemum seticuspe*. The manuscript is generally well-prepared, and the genome will help understand evolutionary of *Chrysanthemum* species and to gene location for useful trait, thus providing reference for genetic basis of important traits in cultivated chrysanthemum. I suggest publication if the following concerns are satisfactorily addressed.

Major comments

1> Autohexaploid or allohexaploid for cultivated chrysanthemum remains controversial. It is not suitable to determine its nature without sufficient data.

Response: Thank you for your suggestion. We have changed instances of 'autohexaploid' to 'hexaploid' in the title and 'segmental allopolyploid' in the abstract and text according to Klie et al., 2014 and Van Geest et al., 2017. The authors of these two studies claimed that cultivated chrysanthemum is a segmental hexaploid, which is also an allohexaploid (originates from interspecific hybrids), but the chromosomes behave like autohexaploids (all six homologous chromosomes are very similar to each other and all homologous chromosomes can form a pair with each other).

2> *Chrysanthemum* is commonly regarded as cultivated chrysanthemum. Thus far there is no evidence for the origination of cultivated chrysanthemum from *C. seticuspe*. It is not persuasive to claim *C. seticuspe* pure line Gojo-0 as a model chrysanthemum.

Response: As you stated, the origin of cultivated chrysanthemum has not yet been

elucidated. However, most researchers have suggested that the *indicum* group species, which includes *C. seticuspe*, are involved in the establishment of cultivated chrysanthemum. Therefore, the genome sequence of *C. seticuspe* is expected to greatly contribute to our understanding of cultivated chrysanthemum traits. *C. seticuspe* shares many traits with the cultivated chrysanthemum, such as photoperiod/temperature-regulated flowering time, flower morphology, seasonal dormancy, , self-incompatibility, etc. Indeed, *C. seticuspe* has been used as a model for cultivated chrysanthemum in Higuchi et al. (2013), Nakano et al. (2019), Hirakawa et al. (2019), and Nakano et al. (2020).

Furthermore, diploid *C. seticuspe* can be hybridized with hexaploidy cultivated chrysanthemum to produce fertile tetraploid chrysanthemum (Taniguchi, unpublished data), suggesting that their genomes are closely related and that direct introgression of *C. seticuspe* genes into cultivated chrysanthemum by crossing is possible. In addition, genes in cultivated chrysanthemum share a high homology with their corresponding genes in *C. seticuspe* (we described it in Supplementary Figure 7 and text in the revised manuscript), suggesting that *C. seticuspe* are a good references for elucidating the function of cultivated chrysanthemum genes. In other words, if a mutant of a particular gene in *C. seticuspe* is identified, one can expect the same phenotype in the cultivated chrysanthemum strain.

We further demonstrated that chromosomes of *C. seticuspe* and the cultivated chrysanthemum (*C. morifolium*) show a clear linear relationship in Supplementary Fig. 14.

We believe that these examples support the chromosome-level, whole-genome sequence of *C. seticuspe* as a useful reference for the analysis and breeding of cultivated chrysanthemum. To make our claim more persuasive, we have described some of the above-mentioned points in the text (mainly L. 322–342).

3> Cultivated chrysanthemum is so high heterozygous that most traits in F1 crosses with contrasting backgrounds showed different genetic variation. The genome of pure line C. seticuspe Gojo-0 is undoubtedly useful, but it difficult to provide refernce for the breeding of heterozygous cultivated chrysanthemums. It should be adequately discussed.

Response: As stated, most loci are heterozygous in cultivated chrysanthemum, which is important for the conventional breeding of cultivated chrysanthemum. But in this manuscript, we have proposed new chrysanthemum breeding strategies such as marker-assisted selection (MAS) and genome editing. We believe that the genome sequence of the pure line Gojo-0 provides useful information for designing DNA makers. DNA markers distinguishing each allele will contribute to MAS even in the heterozygous chrysanthemum. Genome editing can disrupt all six homologous genes, suggesting that the identified recessive homozygous mutant of Gojo-0 can be reconstituted in (heterozygous) hexaploid cultivated chrysanthemum. We have discussed this in the revised text (mainly L368–373, L377-381).

4 > In addition, there are many format errors or typos in manuscript and reference list.

Response: We have addressed these additional minor points indicated in the attached text file and corrected the references. Some changes are mentioned below:

L.2, L.272, L316: ‘autohexaploid’ and ‘autopolyploid’ were revised as described above.

L.54, L56, L309: the words were italicized.

L.77: We added examples.

L. 314: We revised this part as described in Point 2.

L.354: We revised this part as described in Point 3.

REVIEWERS' COMMENTS:

Reviewer #1 (Remarks to the Author):

Although the author has made some changes, there are still some problems with the manuscript. The main problems are as follows.

Line 36: typo: 'aretrotransposon', suggest modification 'a novel retrotransposon family'.

Line 182: The panels were not labelled in Figure 3.

Line 197 - 198: 'There must be autonomous...', must is too strong here, the statement in 195 -197 can only suggest the autonomous element was/is existent.

Line 205 - 211: conflict conclusions about autonomous/nonautonomous SbdRT were stated.

Line 216 - 217: Unproper and disordered abbreviations led to inconvenient reading. For example, before defining the 'nis' in Line 216-217, this phrase has been using a couple of times before.

Line 290: Figure 5a after Gojo-0 is kind confusing, the figure represents AEV02.

Reviewer #2 (Remarks to the Author):

I think most concerns have been satisfactorily addressed. The ms can be accepted for publication.

REVIEWERS' COMMENTS:

Reviewer #1 (Remarks to the Author):

Although the author has made some changes, there are still some problems with the manuscript. The main problems are as follows.

Thank you for your reviewing our manuscript. We revised our manuscript according to your suggestions as follows.

Line 36: typo: 'aretrotransposon', suggest modification 'a novel retrotransposon family'.

We just revised it as 'a retrotransposon family', because Communications Biology does not like to use the word 'novel'.

Line 182: The panels were not labelled in Figure 3.

We added labels in Figure 3.

Line 197 - 198: 'There must be autonomous...', must is too strong here, the statement in 195 -197 can only suggest the autonomous element was/is existent.

We inserted the words 'we speculated that there are'.

Line 205 - 211: conflict conclusions about autonomous/nonautonomous SbdRT were stated.

To clarify our conclusion, we deleted Line 210-213 and added 'In addition, the conserved SbdRT^{nis} copies occupied 52.7% of SbdRT copies (190 out of 360), suggesting that SbdRT was found to be an LTR-RT that is unique as nonautonomous type is highly prosperous (Supplementary Table 3).' At the end of next paragraph (Line 228).

Line 216 - 217: Unproper and disordered abbreviations led to inconvenient reading. For example, before defining the 'nis' in Line 216-217, this phrase has been using a couple of times before.

We revised the order of definition/abbreviation concerning SbdRT-LTR in the section 'Characterization of a novel LTR-RT: SbdRT'.

Line 290: Figure 5a after Gojo-0 is kind confusing, the figure represents AEV02.

We deleted 'which is the ancestor of Gojo-0' in line 294 to avoid confusion.

Reviewer #2 (Remarks to the Author):

I think most concerns have been satisfactorily addressed. The ms can be accepted for publication.

Thank you for your reviewing our manuscript.